# Design of COVID-19 staged alert systems to ensure healthcare capacity with minimal closures

Haoxiang Yang[1], Özge Sürer [2], Daniel Duque[2], David P. Morton [2], Bismark Singh [3], Spencer J. Fox[4], Remy Pasco [5], Kelly Pierce[6], Paul Rathouz[7], Victoria Valencia [7], Zhanwei Du[4], Michael Pignone[7], Mark E. Escott[8], Stephen I. Adler[8], S. Claiborne Johnston[7] & Lauren Ancel Meyers [4,9 ✉]

Community mitigation strategies to combat COVID-19, ranging from healthy hygiene to shelter-in-place orders, exact substantial socioeconomic costs. Judicious implementation and relaxation of restrictions amplify their public health benefits while reducing costs. We derive optimal strategies for toggling between mitigation stages using daily COVID-19 hospital admissions. With public compliance, the policy triggers ensure adequate intensive care unit capacity with high probability while minimizing the duration of strict mitigation measures. In comparison, we show that other sensible COVID-19 staging policies, including France's ICU-based thresholds and a widely adopted indicator for reopening schools and businesses, require overly restrictive measures or trigger strict stages too late to avert catastrophic surges. As proof-of-concept, we describe the optimization and maintenance of the staged alert system that has guided COVID-19 policy in a large US city (Austin, Texas) since May 2020. As cities worldwide face future pandemic waves, our findings provide a robust strategy for tracking COVID-19 hospital admissions as an early indicator of hospital surges and enacting staged measures to ensure integrity of the health system, safety of the health workforce, and public confidence.

---

[1] Los Alamos National Laboratory, Los Alamos, NM, USA. [2] Industrial Engineering and Management Sciences, Northwestern University, Evanston, IL, USA. [3] Department of Mathematics, Friedrich-Alexander-Universität Erlangen-Nürnberg, Erlangen, Germany. [4] Integrative Biology, The University of Texas at Austin, Austin, TX, USA. [5] Operations Research and Industrial Engineering, The University of Texas at Austin, Austin, TX, USA. [6] Texas Advanced Computing Center (TACC), The University of Texas at Austin, Austin, TX, USA. [7] Dell Medical School, The University of Texas at Austin, Austin, TX, USA. [8] The City of Austin, Austin, TX, USA. [9] Santa Fe Institute, Santa Fe, NM, USA. ✉email: laurenmeyers@austin.utexas.edu

Throughout the COVID-19 pandemic, community mitigation activities have proved vital to slowing viral transmission and ensuring the integrity of healthcare systems. Along with viral testing, strategies such as social distancing, face-mask ordinances, business closures, travel restrictions, and stay-home orders have remained paramount, even as safe and efficacious vaccines have become widely available and utilized. By April of 2020, strict stay-home orders were enacted almost universally to combat initial waves of transmission. Within two months, however, many regions lifted restrictions hoping to alleviate socioeconomic hardship, although the risks of resurgence were high[1,2]. In the US—which has reported over 583,000 COVID-19 deaths as of May 13, 2021[3]—communities have scrambled to tighten and relax mitigation policies in response to threatening surges in hospitalizations.

Governmental bodies worldwide have established a variety of COVID-19 alert systems to provide situational awareness and policy directives for the public. They typically monitor one or more data streams—such as COVID-19 case counts, test positivity, hospital capacity, or deaths—and trigger changes in alert level when the data reach specified thresholds[4]. Although most systems include intermediate levels that are intended to slow transmission and reduce the need for full-blown shelter-in-place orders, they vary considerably in complexity, key indicators, and policy levers. France, for example, has a four-stage system; the maximum level is triggered when weekly regional COVID-19 incidence exceeds 250 infections per 100,000 people and COVID-19 patients in the intensive care unit (ICU) occupy at least 60% of capacity[5]. New Zealand, Singapore, South Africa, and the UK have national systems ranging from three to five alert levels that track data on various combinations of COVID-19 incidence, hospitalizations, death, and available healthcare capacity[6,7]; the alert-level system in South Africa includes both economic and public health considerations, while the systems in New Zealand and Singapore focus on public health alone. Within the US, states and cities have established various COVID-19 alert systems. New York's NY Forward Plan[8] monitors seven indicators, including hospital admissions, hospital census, and deaths, to determine a re-opening with four phases. Illinois has a five-stage plan with 11 geographic regions, which tracks test positivity, COVID hospital admissions, and the availability of hospital surge beds, ICU beds, and ventilators[9].

Such COVID-19 alert systems can provide valuable public guidance and flexible policy levers to slow the spread and control alarming surges. However, their public-facing dashboards rarely provide information regarding the underlying design of the system, the choice of data indicators, or the specific thresholds for action. To the best of our knowledge, many are grounded in expert opinion rather than rigorous trade-off analyses that balance COVID-19 burden with economic and social hardship.

Although COVID-19 policies may be dictated by divergent political and cultural considerations, they universally aim to prevent unmanageable surges that threaten the integrity of healthcare systems, like the early pandemic waves in Wuhan, Italy, and New York[10]. Overwhelming numbers of COVID-19 hospitalizations can lead to excess serious complications and mortality for those with COVID-19 or other medical conditions like cancer or cardiovascular disease, who may not receive timely or safe care[11]. For example, influxes of COVID-19 patients have undermined oncology services in the UK[12] and colorectal medicine in Italy[13]. Hospital surges also put healthcare workers at risk, potentially diminishing the workforce and further undermining the quality of care[14,15]. During COVID-19 surges in the US, intensive care units neared capacity more quickly than other medical units[16–18], with trained healthcare professionals rather than space, medical equipment, or PPE being the key limiting resource[17]. Early data from China and Italy suggest that 5% of cases who test positive for COVID-19[19] and 16% of hospitalized patients require ICU-level care[20]. In the fall and winter of 2020-2021, cities throughout the US and Europe again faced overwhelming COVID-19 healthcare surges despite community mitigation efforts[21,22]. Many have deployed temporary medical facilities, often called field hospitals or alternate care sites, to accommodate overflow, although most are not equipped to provide ICU care[23]. For example, the Javits New York Medical Station has 42 ventilators and the Navy hospital ship USNS Comfort has 100 ICU beds as of April 2020[24], and Wisconsin, with over 85% of the state's hospital beds and over 88% of the state's ICU beds occupied in early November 2020[25], began sending patients to a field hospital at the Wisconsin State Fair Park[26].

In this study, we apply stochastic optimization to recommend policy triggers governing stages of community mitigation to prevent overwhelming hospital surges and ensure adequate capacity in the unlikely case that they occur. Strict community mitigation measures, such as shelter-in-place orders, are socioeconomically detrimental and only proposed when the existing healthcare system risks inundation. Our data-driven optimization model is built atop a high fidelity SEIR-style (susceptible-exposed-infectious-recovered) simulation model of SARS-CoV-2 transmission. We can rapidly solve for optimized thresholds for daily COVID-19 hospital admissions at which community mitigation measures should be enhanced or relaxed. To validate the approach, we compare the optimized policies to established policies in terms of the expected duration of restrictive closures and the probability that COVID-19 will overwhelm local healthcare capacity.

Our principled framework can guide public policy, reducing socioeconomic hardship while ensuring the integrity of the healthcare system. Our framework was rapidly developed and applied by a task force of scientists, public health authorities, hospital systems, and elected officials during April and May of 2020 to create a robust COVID-19 alert system that has been used for nearly a year to guide public policy in the Austin, Texas metropolitan area, with a population of about 2.2 million[27]. Here, we significantly extend a previously published pilot study[28], which toggled between just two stages, into a practical data-driven framework for building staged alert systems to mitigate competing risks in the face uncertainty and provide actionable policy insights based on the experiences in Austin. To demonstrate the versatility of the method, we optimize a similar alert system for the larger Houston, Texas MSA, and extend the method to stand up an alternate care site if mitigation measures fail (see Supplementary Discussion 1).

## Results

To solve for optimal policies, we simulate COVID-19 transmission under a staged alert system using a stochastic SEIR model, which includes ten compartments for each of ten age-risk categories. In the model, the alert stages govern the COVID-19 transmission rate and the stages change when the seven-day moving average in COVID-19 hospital admissions crosses defined thresholds. Our stochastic optimization model identifies stage-specific thresholds that minimize the total expected cost while ensuring sufficient healthcare capacity with high probability, using Monte Carlo estimates. We provide results for the Austin, Texas MSA, which implemented a COVID-19 alert system built via this approach on May 13, 2020[29], and provide analogous results for the Houston, Texas MSA in Supplementary Discussion 1.

Our model includes four alert stages, blue (new normal), yellow (moderate risk), orange (high risk), and red (very high risk), that

progressively reduce transmission from an unmitigated baseline, and more so among high-risk sub-populations (Table 1). The reductions are based on the least-squares fitting of the model to comprehensive COVID-19 hospitalization data from the Austin MSA from February 28 through October 7, 2020 (Supplementary Method 2).

During the June 2020 pandemic surge the three major hospital systems in Austin estimated a total COVID-19 inpatient capacity of 1500 beds, including a COVID ICU capacity of 331 beds. Based on COVID-19 hospitalization and ICU counts in Austin, we estimate that the daily proportion of COVID-19 patients in ICUs dropped from 45% to 30% from March 19, 2020 to August 10, 2020. Even at 30%, ICU capacity would likely be breached before general hospital capacity, as was corroborated by the June 2020 COVID-19 surge (Fig. 1). Thus, we design policies to ensure that COVID-19 healthcare demand does not exceed the tighter constraint of ICU beds. While general ward beds can be converted to ICU beds, the requisite critical care nurses, physicians, and equipment such as ventilators can be in short supply precisely when needed.

Based on 12-month COVID-19 projections for Austin starting on October 7, 2020, we identify thresholds that provide at least 95% assurance that Austin will not run out of ICU capacity while minimizing the overall socioeconomic cost, represented by a sum of daily penalties whose magnitude grows with stricter stages of mitigation. Increasingly strict stages of yellow, orange, and red are triggered when the rolling seven-day average of COVID admissions exceeds 10, 20, or 120 cases, respectively (Table 2). Assuming the observed reduction in transmission from July until October 7th, we expect that hospitalizations will rise to the point of triggering the orange stage by November and possibly requiring a short-lived lock-down (red) between late November and mid-March (Fig. 1).

We compare the optimized triggers to four alternative policies—an optimized two-stage system (with access only to the red and yellow stages) that again respects ICU capacity, an optimized four-stage system that instead ensures total hospital capacity is respected with high probability (0.95), thresholds based on France's COVID-19 alert system, and widely cited reopening criteria developed by the Harvard Global Health Institute (HGHI)[30] (Table 2 and Fig. 2). The policy optimized to preserve overall inpatient rather than ICU capacity fails to ensure safe ICU capacity with an estimated 20% chance of an unmanageable surge. The distribution of ICU patient-days above capacity is highly skewed with a median of 0 days, a 95th percentile of 1476 days, and a 99th percentile of 2388 days. Likewise, the France-based policy has a 38% chance of exceeding ICU capacity with the median, 95th, and 99th percentiles of 0, 1273, and 2948 days. The other two policies err on the side of

---

**Table 1 Structure and impact of a four-stage COVID-19 alert system.**

| Stages | Example measures | Transmission reduction | $R_t$ |
|---|---|---|---|
| Red | Shelter-in-place order: mask mandate, no public activities, gatherings, or travel | Largest (78.2%) | 1.02 |
| Orange | Mask mandate, no indoor dining, no medium or large gatherings | Moderate (69.2%) | 1.45 |
| Yellow | Mask mandate, partial limitations on indoor dining and bars, no large gatherings | Modest (60.3%) | 1.87 |
| Blue | New normal: avoid large gatherings, masks and physical distancing recommended | Lowest (51.3%) | 2.29 |

Colors indicate stages. For each stage, the table provides example measures, which may evolve with future data on the impact of mitigation strategies and roll-out of surveillance testing. Transmission reduction estimates and reproduction numbers are derived from COVID-19 hospital admissions data from the Austin, Texas MSA during a period that included a stay-home order, a re-opening phase that led to an early summer surge, followed by reduced transmission with the implementation of face-mask requirements and reinstatement of other distancing measures, and an uptick in spread as fall began (February 18 to October 7, 2020). To allow comparison, reproduction number estimates are given relative to a fully susceptible population. The model assumes high-risk sub-populations are sheltered to a greater degree, as detailed in the Supplementary Method 2.

---

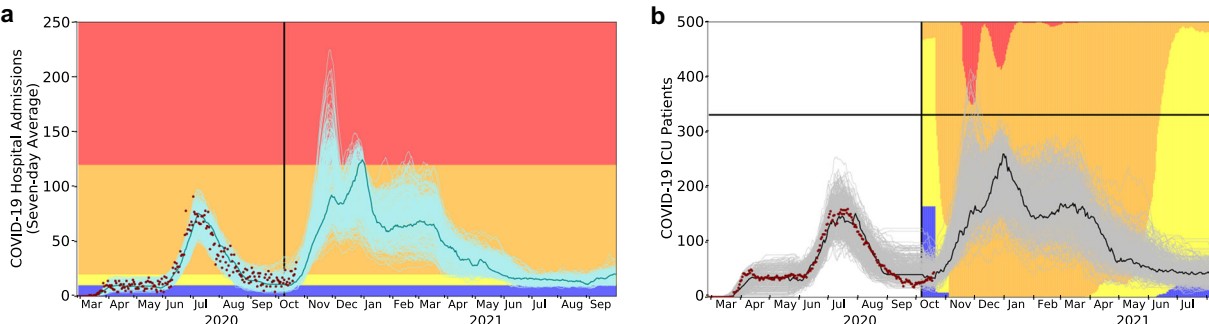

**Fig. 1 COVID-19 healthcare projections for Austin under the optimized staging policy, from October 7, 2020 through September 30, 2021.** The strategy was derived to minimize the expected days in costly alert stages while respecting intensive care unit (ICU) capacity. In both plots, the light curves indicate 300 stochastic simulations, the single solid curve is a representative central projection, the red points correspond to the reported COVID-19 admissions and ICU census for all Austin area hospitals through October 20, 2020, and the vertical black line indicates the start of the projection period. **a** The COVID-19 alert level changes when the seven-day moving average of daily COVID-19 hospital admissions crosses optimized thresholds, as indicated by the colored horizontal bands for the red, orange, yellow, and blue alert stages. **b** The policy provides a 95% guarantee that the number of COVID-19 ICU patients does not exceed the estimated local capacity of 331 beds (black horizontal line). The background colors represent the proportion of the simulated scenarios in each alert stage on each day. For example, on January 1, 2021, 15% of projections are in the most restrictive red stage, and the remaining 85% are in the orange stage.

**Table 2 Performance comparison across five COVID-19 staging policies.**

| Indicator data | Policies | | | | |
| --- | --- | --- | --- | --- | --- |
| | COVID-19 hospital admissions (7-day average) | | | Percent ICU (France) | Incidence (Harvard) |
| | Optimal (ICU capacity) | Optimal two-stage (ICU capacity) | Optimal hospital (overall capacity) | Percent ICU beds occupied by COVID-19 | New cases per 100 000 (7-day average) |
| Thresholds | | | | | |
| blue (low risk) | <10 | — | <10 | — | <1 |
| yellow (moderate risk) | 10–20 | <90 | 10–20 | <30% | 1–10 |
| orange (high risk) | 20–120 | — | 20–200 | 30%–60% | 10–25 |
| red (very high risk) | >120 | >90 | >200 | >60% | >25 |
| Median days in red stage [90% PI] | 14 [0–16] | 65 [47–78] | 0 [0–0] | 39 [23–55] | 42 [28–71] |
| Probability ICU demand exceeds capacity | 2.7% | 1.7% | 20.0% | 38.0% | 0.0% |
| Median peak ICU demand (patients) | 255 | 268 | 275 | 312 | 122 |
| 95th percentile of peak ICU demand | 317 | 309 | 412 | 426 | 157 |
| Median unserved ICU demand (patient–days) [90% PI] | 0 [0–0] | 0 [0–0] | 0 [0–1476] | 0 [0–1273] | 0 [0–0] |
| 99th percentile of unserved ICU demand (patient-days) | 374 | 9 | 2388 | 2948 | 0 |

From left to right, the three optimal policies are derived to prevent overwhelming ICU demand using either a four-level or two-level alert system or to prevent overwhelming inpatient demand using a four-level system. As benchmarks, we evaluate policies implemented in France[5] and proposed as gating criteria for relaxing measures and opening schools[30]. For Austin, the orange and red thresholds for the Percent ICU (France) policy translate to 99 and 199 COVID-19 ICU cases, respectively; the yellow, orange, and red thresholds for the Incidence (Harvard) policy translate to 220, 2200, and 5500 new cases, respectively, assuming that one in ten cases is reported. We implemented each policy in our stochastic SEIR model fit to hospitalization data for the Austin MSA, assuming the reported COVID-19 ICU and inpatient capacities of 331 and 1500 beds, respectively. Outcomes are based on 300 stochastic simulations of COVID-19 transmission and healthcare burden from October 7, 2020 through September 30, 2021 under each policy.

caution, providing strong guarantees that COVID-19 will not overwhelm ICU capacity, but at the cost of longer duration of lockdowns. Under the optimal strategy, we would expect a median scenario to have two weeks of stage-red restrictions (14 [90% PI: 0–16] days). The median lock-down periods increase to nine weeks and six weeks under the optimized two-stage system and Harvard-based policy, respectively.

## Discussion

As US states relaxed and reinstated community mitigation measures during the early months of the COVID-19 pandemic, policymakers sought clarity on which data to track and when to take action. In April of 2020, we developed this data-driven optimization framework out of necessity, as the civic, healthcare, and public health leadership in the Austin metropolitan area raced to implement a robust policy that would ensure the integrity of area hospitals while minimizing socioeconomic damage and complying with state-mandated reopening orders. The solution we derived for Austin, which continues to guide policy as of May 2021, is to track daily new COVID-19 hospital admissions as an early indicator of hospital surges and enact staged restrictions when the 7-day moving average crosses predetermined thresholds[27,31]. For plausible COVID-19 scenarios in Austin (Results) and Houston (Supplementary Discussion 1), we find that limited stay-at-home (red) periods should suffice to respect healthcare capacity.

In developing this approach, we addressed two early policy challenges that still persist in many jurisdictions. The first is identifying a source of data that provides reliable and timely COVID-19 situational awareness. The two most widely collected and cited indicators—case counts and death counts—give unreliable signals. Spikes and dips in confirmed case counts reflect the rapidly changing capacity and purview for testing, perhaps more than the pace of the pandemic itself, which may be compounded by long turn-around times[32,33] and delayed data reporting due to aged IT infrastructure and an over-tasked workforce[34,35]. Setting aside possible under-reporting of deaths, COVID-19 mortality data should more clearly indicate whether policies are having the desired effect on repressing transmission, but not until several weeks after the fact, given the 3-week average course of fatal disease, and given the time taken to report such deaths. COVID-19 hospitalization counts may provide similar fidelity with a shorter lag, but inferring transmission rates from such data may be complicated by variable duration of hospital stays, depending on the changing demographics of COVID-19 cases and the availability of alternative post acute-care facilities. Of the various data streams, we find that COVID-19 hospital admissions provide the clearest early signal of recent transmission and imminent hospital surges. However, hospital admissions are not typically reported on city, county, state, or national COVID-19 dashboards. Austin's efforts to collect and publicize this data[27] serve as an exemplar of local leadership providing decision-makers and the public with a reliable real-time indicator of changing pandemic risks.

The second persistent challenge is articulating clear policy goals that reflect the universal desire to prevent suffering and loss of both life and livelihood while ensuring consistency with state and federal requirements. Early deliberations led Austin's leadership to posit a two-part goal. The first goal is to prevent overwhelming surges in hospitalizations that would potentially increase morbidity and mortality for any patients requiring care and increase risks to the healthcare workforce. The second goal is to minimize the duration of economically restrictive policies. Designing policies to achieve concise and widely acceptable objectives, like these, allows policymakers to anticipate potential

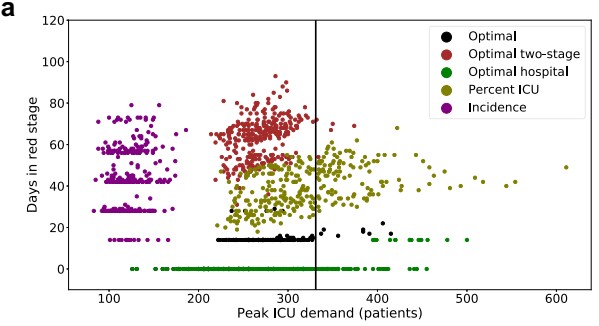
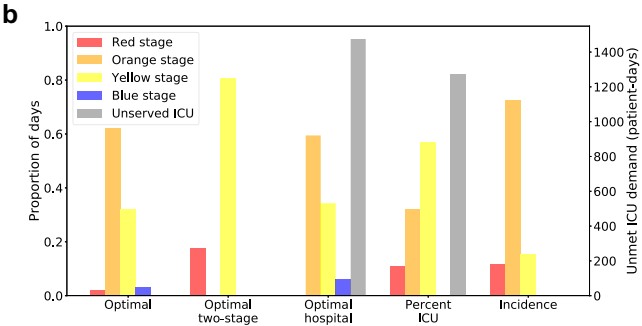

**Fig. 2 Projected intensive care unit (ICU) surges and days under lock-down for the optimized strategy versus four alternative strategies.** Optimal is the recommended strategy. The Optimal two-stage strategy is optimized to respect ICU capacity under a two-stage alert system, and the Optimal hospital strategy respects total hospital capacity under a four-stage alert system. The Percent ICU strategy is based on France's mitigation policy[5] and the Incidence strategy is based on reopening criteria proposed by the Harvard Global Health Institute[30]. **a** The maximum daily number of COVID-19 patients in ICUs versus the number of days under the most restrictive red alert level. Each point represents the result of a single stochastic simulation under one of the five policies (indicated by color). The plot includes 300 points per policy; the vertical black line indicates the estimated COVID-19 ICU capacity of 331 beds for the Austin area. The vertical stratification of the green and purple points stems from a model assumption that stages must be in place for a minimum of 14-days before they can shift. The Optimal policy is designed to minimize the use of costly stages while having 95% of the peak-demand values within ICU capacity. On the rare occasion that hospital admissions transiently exceed the red threshold, a return to orange is often triggered as soon as the 2-week minimum passes. **b** The expected proportion of days spent in each stage, colored in the same manner as Fig. 1 and the 95th percentile of unmet ICU demand measured in patient-days above capacity, in gray with values indicated on the right y-axis.

limitations and provide transparent and intuitive justifications for the public.

Cities, states, and countries worldwide have enacted staged COVID-19 mitigation policies. However, few provide detailed rationales for the choice of data indicators or the trigger conditions for changing stages. They may, in fact, be grounded in rigorous assessments of both data reliability and probability of achieving explicit policy goals. However, our comparative evaluation of two examples—France's ICU-capacity-based triggers and the Harvard Global Health Institute's incidence-based triggers—suggest that such systems may be sensible but sub-optimal. They may err in the direction of either failing to prevent overwhelming surges of COVID-19 hospitalizations or imposing unnecessarily early or long restrictions.

Throughout the COVID-19 pandemic, various branches of the US government, including city and state authorities, have engaged in highly polarized disputes regarding COVID-19 mitigation[36]. We designed this optimization framework during the spring of 2020, amidst considerable national tension over the White House's Opening Up America Again plan[37]. In that climate, we aimed to provide an adaptive decision framework that would be universally acceptable to all stakeholders, including our city, state, and federal governments, to ensure that the policies would not face political or legal challenges. Given the horrific images from hospitals in Italy and New York in early 2020[38,39], there was broad consensus that local authorities should take measures to prevent overwhelming healthcare surges while opening up the economy as much as possible. This led us to align our alert systems towards two goals—ensuring hospital capacity is not overrun while minimizing the duration of restrictive measures.

Since Austin implemented the recommended staged alert system in May 2020, city leadership has proactively socialized the framework through a public-facing dashboard that tracks hospital admissions and visualizes the key thresholds[27], and through daily public messaging via news outlets and social media[40]. Behind the scenes, the city's COVID-19 task force has continually pressure tested and updated the alert system, as our understanding of the virus, local healthcare resources, and behavioral responses have changed. For example, in October 2020, the major hospital systems reduced their estimate of COVID-19 ICU capacity from 331

beds to 200 beds, stemming from an increase in non-COVID patients and staffing challenges. We quickly updated our optimization analysis and determined that the triggers for transitioning to the strictest orange and red stages should be reduced. However, to avoid undermining public trust, the city did not immediately announce the policy change. Instead, they waited until hospitalizations began trending upwards, but with ample time to cultivate community buy-in before the triggers hit. In accordance with the revised triggers, Austin transitioned to the most stringent alert stage (red) on December 23, 2020, and relaxed to the orange and then yellow stages on February 9, 2021, and March 13, 2021, respectively, when hospital admissions dropped below the corresponding thresholds.

Austin's staged system was optimized to prevent catastrophic healthcare surges while minimizing the duration of costly measures. As designed, the shift to red in December prevented an overwhelming surge in COVID-19 ICU utilization, but just barely. The ICU census in the metropolitan area peaked just below the local capacity of 200 beds on January 12, 2021 (Supplementary Discussion 2). As an indirect byproduct of flattening the hospitalization curve, the system has also mitigated overall morbidity and mortality. As of March 22, 2021, Travis County (Austin) reported 73 COVID-19 deaths per 100,000 residents, which was considerably lower than the statewide death rate of 161 per 100,000[41]. Other major metropolitan areas in Texas fared worse, with Harris County (Houston), Dallas County (Dallas), Bexar County (San Antonio) reporting 120, 143, and 162 COVID-19 deaths per 100,000 people, respectively. Texas' hardest hit regions include the Rio Grande Valley with 281 (Hidalgo County) and 335 (Cameron County) COVID-19 deaths per 100,000 people, and West Texas with 300 (El Paso) and 249 (Lubbock) COVID deaths per 100,000 people. The staged alert system also seems to have achieved the goal of reducing socioeconomic costs. Across 22 different trauma service areas of Texas[42], the Austin area spent the fewest days under state-ordered restrictions on elective surgeries and restaurant/bar/retail occupancy during the winter surge (Executive Order G32)[43].

There are important limitations to our approach. If the reporting of hospitalization data is delayed, biased, or inconsistent, the system may be prone to false or delayed alarms[44].

Second, our analyses make strong assumptions about the reduction in transmission under various alert stages. Although these are directly estimated from data during various pandemic periods in Austin, the future impact of staged restrictions may change with public willingness to comply with guidelines on distancing, face coverings, and hygiene. If future restrictions lead to greater reductions in transmission than assumed in our model, then the policies would be conservative. That is, their guarantee against overwhelming ICU surges would exceed 95%, at the expense of longer than necessary restrictions. If behavior is more lax than expected, then the policy may fail to protect the healthcare system, potentially requiring personnel and resources from state or federal agencies, transferring of patients to other jurisdictions, or use of alternate care sites. Third, our model assumes that hospital and ICU capacity for COVID-19 patients is fixed. However, other events or diseases, such as natural disasters or seasonal influenza may cause substantial reductions in COVID-19 capacity. While predictable fluctuations can be incorporated into the model a priori, unpredictable events may pose significant risks. In fact, we have been validating our model assumptions on a weekly basis throughout the pandemic. Our Austin COVID-19 healthcare dashboard[45] provides daily estimates of the current reproduction number and three-week out projections for hospital and ICU demand. As behavior and conditions in the city have evolved, we have adjusted our model and re-derived the optimal trigger policy thresholds. In all but one case, we determined that the originally derived triggers were robust even if no longer optimal per our model. The one exception was an October 2020 update of local ICU capacity, based on increased occupancy by non-COVID cases, that suggests a need to trigger the orange and red stages earlier than originally prescribed.

Our framework for designing policy provides a path for any city to reduce the need for strict shelter-in-place orders while ensuring the integrity of the health system, safety of the health workforce, and public confidence. It can be flexibly tailored to determine policy triggers based on local demographics, health risks, behavioral responses to COVID-19, and healthcare capacity, as demonstrated for the city of Houston, Texas (see Supplementary Discussion 1). Describing the strategy is easy—track daily new COVID-19 hospital admissions as a reliable indicator of an impending surge in hospitalizations and trigger changes in policy when the seven-day moving average crosses predetermined thresholds. Implementation of this strategy is harder but can be done quickly with little additional cost. It requires adding daily hospital admission counts to COVID-19 dashboards, firming up estimates for local COVID-19 ICU hospital surge capacity, and using straightforward models to determine early warning indicators for when a surge is coming.

As of May of 2021, SARS-CoV-2 vaccines are rolling out unevenly across the globe and the virus continues to wreak havoc on several continents. The continual emergence of new viral variants may cause future resurgences, even in highly vaccinated communities. Thus, staged alert systems will continue to be an important option for mitigating the risks of COVID-19, as well as future pandemics, particularly if they are carefully tailored to the changing state of the pandemic and effectively communicated to encourage public compliance.

## Methods

Our SEIR simulation model assumes that pre-symptomatic, symptomatic, and asymptomatic cases have different levels of infectiousness. Contact rates vary by age group, differ on weekdays and weekends, and decrease during school holidays and closures. We account for micro-stochastics using binomially distributed numbers of transitions between compartments and macro-stochastics by randomly sampling parameter values from specified distributions; see Supplementary Methods 2, 3. We train, i.e., optimize, a policy on one set of 300 Monte Carlo

simulations and require 95% stay within capacity, but report results using 300 independent, i.e., out-of-sample, test scenarios. Supplementary Method 1 details the optimization model and its probabilistic constraint. The prediction intervals and the percentiles we report are based on order statistics from the 300 test scenarios. The simulation and optimization models and the optimization algorithm are implemented in Python.

We model four alert stages, each corresponding to a different level of transmission reduction. A day spent in the blue stage incurs a unit socioeconomic cost, and each day spent in the next most restrictive stage increases the cost by a factor of 10. The trigger policy works as follows: as hospitalizations rise, we move to a stricter stage of physical distancing when the seven-day moving average of daily admissions surpasses a threshold. As hospitalizations fall, we relax restrictions when the seven-day moving average falls below the same threshold. These rules are coupled with a requirement that we spend at least two weeks in a stage.

Institutional Review Board (IRB) approval was not required for two reasons. First, the implementation of the staged alert system was deemed to be public health practice, and hence not subject to IRB approval[46]. Second, hospitalization admissions and census data used to inform the staged alert system are publicly available[27], and the use of anonymized patient experience data (Supplementary Method 2) was approved as exempt human subjects research by the IRB of The University of Texas at Austin.

**Reporting summary**. Further information on research design is available in the Nature Research Reporting Summary linked to this article.

## Data availability

The hospital admissions and census data that supported this study can be found at the GitHub repository: https://github.com/haoxiangyang89/COVID_Staged_Alert. The remaining data used in our analysis are also available at the GitHub repository and further made available and described in the Supplementary Information.

## Code availability

The codes for performing the required analysis, and detailed instructions on installing and running the codes, are available at the repository https://github.com/haoxiangyang89/COVID_Staged_Alert. The DOI for the GitHub repository is https://doi.org/10.5281/zenodo.4759278[47].

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

## Acknowledgements

The authors thank Achyut Kasi and Cindy Sanchez for code to produce graphs and reports used in the manuscript. The Center for Nonlinear Studies at Los Alamos National Laboratory supported Haoxiang Yang's work. Bismark Singh was co-financed by the Bavarian-Czech Academic Agency with funds from the Free State of Bavaria. This work was further supported by the National Institutes of Health under Grant NIH R01 AI151176 and by the U.S. Department of Homeland Security under Grant 2017-ST-061-QA0001. The views and conclusions contained in this document are those of the authors and should not be interpreted as necessarily representing the official policies, either expressed or implied, of the U.S. Department of Homeland Security.

## Author contributions

H.Y., D.P.M., and L.A.M. designed the research; H.Y., Ö.S., D.D., D.P.M., P.R., Z.D., and L.A.M. performed the research; H.Y., Ö.S., D.P.M., R.P., K.P., P.R., V.V., and L.A.M. analyzed data; M.P., M.E.E., S.I.A., and S.C.J. provided valuable insights regarding public health policy and socializing the trigger-based staged-alert system; and H.Y., Ö.S., D.P.M., B.S., S.J.F., Z.D., and L.A.M. wrote the paper.

## Competing interests

The authors declare no competing interests.
