## [Peer Review File · Nature Communications]

Reviewers' comments:

Reviewer #1 (Remarks to the Author):

This paper describes an implementation of the method proposed by Duque et al. (2020, Timing social distancing to avert unmanageable COVID-19 hospital surges. Proc Natl Acad Sci USA 117, 19873–19878). The core idea is to use COVID-19 hospital admissions as a leading indicator of hospital surges to guide the transition between different mitigation measures (i.e., stay-at-home orders, mandated masking, indoor dining limitations, and restrictions of large gatherings). The authors apply the core idea to the case of Austin, Texas and demonstrate that their method leads to a sensible tool for guiding COVID-19 policymaking and “a path for any city to reduce the need for strict shelter-in-place orders while ensuring the integrity of the health system, safety of the health workforce, and public confidence.”

The paper is well written and the supplementary information is detailed and informative. In general, I feel this work deserves some consideration for publication in the journal.

That being said, I have serious reservations about its novelty given the existence of Duque et al. (2020), which presents essentially the same strategy — use hospital admissions as the trigger to switch between different mitigation measures. It is true that the current manuscript is more comprehensive and uses more up-to-date data, but the core idea remains largely the same and does not offer novel insights for health policymakers. Thus, I feel a more specialized journal (e.g., Clinical Infectious Diseases) may be a better fit.

In addition, whether optimization model (3) in the supplementary information is a correct representation of a policymaker’s priorities is debatable. The objective (3a) seeks to minimize the total socioeconomic costs associated with each of the mitigation strategies. What about infections and mortality? Clearly, the solution to this model is sensitive to the cost parameters C_1 , C_2 , C_3 , and C_4 . In Supplementary Table 10, the authors wrote that $C_1 = 10000$, $C_2 = 100$, $C_3 = 10$, and $C_4 = 1$ without justifying their choices of such parameters values. These choices seem rather arbitrary and significantly weakens the credibility of the model and its recommendations.

Reviewer #2 (Remarks to the Author):

The authors present a detailed, data-driven approach to defining triggers for the implementation and relaxation of restriction policies aimed at combating the spread of COVID-19 and maintaining hospital ICU capacity at a level appropriate for treatment of critical cases, during the second wave of COVID-19 in Austin and Houston, TX.

I like this paper, and appreciate the transparent, principled approach the authors took. I am particularly pleased that the authors chose to submit this framework for publication, as it was used to inform policy and therefore will maintain a record of policy decision support methods.

My principal critique is that the method and qualitative results bear strong resemblance to those presented in the precursor paper published in PNAS.

Therefore, this paper would benefit from an explicit discussion of how this work builds upon the previous effort, and specifically how the policy recommendations changed. To be of sufficient novelty for publication in this venue, it is my opinion that some fundamental difference or major scientific development should differentiate the two iterations of the work. Currently it is unclear whether such a difference exists. I say it is unclear because the model, the optimisation procedure, and the calibration procedure are all of a nuanced and rigorous nature and for a reader not intimately acquainted with the research, it is difficult to ascertain whether any of the extensions to the proof-of-concept constitute a fundamental step of scientific importance.

The developments in this work are clearly of operational importance to policy design (i.e., replacing ad-hoc parameter estimates with data-driven ones). If the main contribution is the rigorous application of the method described earlier to the second wave of COVID-19 in Austin and

Houston, then this relationship between the two pieces of work should be clearly emphasised, and a convincing case made for the importance of that contribution to the scientific literature.

In addition to this general critique, I have several more minor questions and comments that I believe could improve the manuscript regardless of where it is published:

1) clarity - currently, both rigorous quantitative, and verbal descriptions of the modelling choices and methods exist in the supplemental material, and I appreciate this greatly. However, clarity would be improved by adjusting the organisation of the supplemental text, tables, and figures, so that the qualitative, verbal descriptions consistently precede the rigorous mathematical descriptions. As a naïve reader, it took quite a while for me to put together all of the pieces, from model, through parameter calibration, optimisation, and sensitivity analysis. I understand that the methods are elaborate, and determining the best presentation is not an easy task, but the current presentation could be improved significantly (though I believe all of the pieces are there).

2) stochastics and scenario rejection - it would be very useful to understand the role of macro- vs micro-stochastic effects in determining the 'accept' or 'reject' decision described in supplemental section C (selection of scenarios). While I understand that describing this exhaustively may be out of scope, perhaps the overall rejection rate could be provided, along with a rejection rate for fixed macro-stochastics (i.e., those corresponding to the 'representative trajectory') where only micro-stochastics produce rejection. I think this particular monte carlo method of forecasting is interesting (I have not encountered it before) and understanding the rejection decision in more detail would be beneficial, as it appears to be useful in providing robust predictions and recommendations.

3) contemporary situation - based on my examination of recent (2nd wave response) events in the Austin area, the thresholds discussed in this piece do not correspond precisely to practice. With the benefit of hindsight, perhaps the authors could add some discussion as to how the situation changed since this work was first produced - which assumptions and modelling decisions have held up, and which have not? This could be added to the discussion ending on line 192 (re: ICU capacity in October).

Overall, this work provides a refreshingly thorough description of what was clearly a rapidly produced system for emergency response. I support publication of the manuscript pending further differentiation from the proof-of-concept piece published by Duque et al. (PNAS, 2020). In addition, any effort to address the minor comments listed above would be appreciated.

Cameron Zachreson

Response to Referees, NCOMMS-20-48848: Timing Social Distancing to Avert Unmanageable COVID-19 Hospital Surges

Referee #1:

Comments:

This paper describes an implementation of the method proposed by Duque et al. (2020, Timing social distancing to avert unmanageable COVID-19 hospital surges. Proc Natl Acad Sci USA 117, 19873–19878). The core idea is to use COVID-19 hospital admissions as a leading indicator of hospital surges to guide the transition between different mitigation measures (i.e., stay-at-home orders, mandated masking, indoor dining limitations, and restrictions of large gatherings). The authors apply the core idea to the case of Austin, Texas and demonstrate that their method leads to a sensible tool for guiding COVID-19 policymaking and “a path for any city to reduce the need for strict shelter-in-place orders while ensuring the integrity of the health system, safety of the health workforce, and public confidence.”

The paper is well written and the supplementary information is detailed and informative. In general, I feel this work deserves some consideration for publication in the journal.

That being said, I have serious reservations about its novelty given the existence of Duque et al. (2020), which presents essentially the same strategy — use hospital admissions as the trigger to switch between different mitigation measures. It is true that the current manuscript is more comprehensive and uses more up-to-date data, but the core idea remains largely the same and does not offer novel insights for health policymakers. Thus, I feel a more specialized journal (e.g., Clinical Infectious Diseases) may be a better fit.

We appreciate this concern and acknowledge that our original submission did not sufficiently distinguish the current work from Duque et al. (2020). We have substantially revised the manuscript and supplement to highlight its novelty and distinct methodological advances, scientific insights and public health applications, including:

1. Actionable policy insights and benchmarks:

The authors of this study have worked together to develop, validate, and consistently follow the staged alert system that has been guiding the COVID-19 response in the City of Austin, Texas since May of 2020; see the Austin Staging Dashboard (Link). Our earlier work in Duque et al. (2020) pre-dated the implementation of this system as well as the summer 2020 COVID-19 surge, both of which substantially changed our understanding of the virus and the challenges faced by city leaderships and healthcare systems. Since Duque et al. (2020), we substantially revised and extended the model to (i) improve fidelity, (ii) derive an actionable policy that achieves the city’s health and economic goals and allows consistent public messaging to cultivate community buy-in, and (iii) enable continual validation and rapid analysis of emerging policy challenges (such as triggering the launch of a field hospital). We have revised the manuscript to more fully describe the context and impact of the model on local decision making, public awareness, and epidemiological outcomes. In addition, we explicitly compare Austin’s *optimized* triggers to case-count thresholds recommended by Harvard Global Health and ICU-based thresholds used to trigger lock-downs in France, to provide policy insights that are applicable to COVID-19 control in cities worldwide.

2. Methodological advances:

In our current manuscript, the optimization model, i.e., Eqs. [3] in the supporting information (SI) document, provides significant advances relative to Duque et al. to facilitate the analyses and insights referenced above. Rather than toggling between two alert levels as in Duque et al., the current manuscript is based on the five alert stages in use in Austin, and many jurisdictions, and we further extend the model to distinguish ICU capacity,

general ward capacity, and surge capacity. (The most relaxed “new normal” stage is not used in our analysis and so we simplify the presentation to four stages.) This required methodological advances. First, we developed a means to directly estimate parameters governing the impact of control measures from local hospitalization data in Austin and Houston. Second, we developed a new Monte Carlo procedure to infer key model parameters to help ensure our model is consistent with observed hospitalization data. Duque et al. did not analyze hospitalization data to inform parameters for different control measures. These nontrivial methodological advances were essential to deriving, validating, and maintaining a robust multi-staged alert system in Austin since May 2020, and will allow flexible application of the framework for managing future COVID-19 and other pathogen risks in cities worldwide.

3. *Use in practice:*

Since we originally submitted this work for publication in *Nature Communications* in December, the city of Austin has continued to follow the optimized staged alert system described in our manuscript. As prescribed by the policy, Austin declared a transition to the red alert stage when the seven-day average COVID-19 hospitalizations surpassed the specified threshold on December 23, 2020 and relaxed back to the orange state when the admissions declined below the threshold on February 9, 2021. The policy worked as designed. First, the COVID-19 ICU usage peaked just below the local capacity on January 12th. Second, across the 22 Trauma Service Areas of Texas, the Austin region spent the fewest days under state-ordered restrictions on elective surgeries and restaurant/bar/retail occupancy (Executive Order G32). Third, as of March 22, 2021, Texas reported 161 COVID-19 deaths per 100,000 people statewide. In Harris County (Houston), Dallas County (Dallas), Bexar County (San Antonio), the estimated tolls were 120, 143, and 162 COVID-19 deaths per 100,000 people, respectively. Some of our hardest hit regions include the Rio Grande Valley with 281 (Hidalgo County) and 335 (Cameron County) COVID-19 deaths per 100,000 people, and West Texas with 300 (El Paso) and 249 (Lubbock) COVID deaths per 100,000 people. In contrast, Travis County (Austin) reported 73 per 100,000 people during the period, and the five-county Austin MSA reported 77.

We expand on these points in the Appendix below.

In addition, whether optimization model [3] in the supplementary information is a correct representation of a policy-maker’s priorities is debatable. The objective [3a] seeks to minimize the total socioeconomic costs associated with each of the mitigation strategies. What about infections and mortality? Clearly, the solution to this model is sensitive to the cost parameters C_1 , C_2 , C_3 , and C_4 . In Supplementary Table 10, the authors wrote that $C_1 = 10000$, $C_2 = 100$, $C_3 = 10$, and $C_4 = 1$ without justifying their choices of such parameters values. These choices seem rather arbitrary and significantly weakens the credibility of the model and its recommendations.

1. In April of 2020, our team of scientists, public health professionals, and policy makers carefully deliberated the choice of objective functions and considered a wide range of outcome measures, including infections, mortality, and various socioeconomic indicators. At the time, the state governor had indicated that he would mandate a two-staged reopening of Texas, starting in May of 2020, and would not allow cities to maintain strict shelter-in-place orders. Moreover, an April 2020 executive order prevented local governments from enacting face-mask requirements. In that climate, we aimed to identify public health and socioeconomic objectives that would be acceptable to all stakeholders, including the state government, to ensure that the policies would not face political or legal challenges. Optimizing to minimize infections and mortality yields policy recommendations with extended lock-down periods that would have put the city in conflict with the state. However, given the horrific images from hospitals in Italy and New York in spring 2020, there was broad consensus that local authorities should take measures to prevent overwhelming healthcare surges (which we implement as a probabilistic constraint) while opening up the economy as much as possible. This led us to focus on hospital capacity and specify a tiered cost function that strongly penalizes the two strictest alert stages. We explored multiple monotonic objective function penalties, and given the context just sketched, we found what we regard as an appropriate—and easily interpreted—objective to which the triggers are relatively insensitive. (See item 2 below.)

In fact, the city took a bit of a gamble in publicizing the alert stages in late May 2020. It was unclear whether the state would overrule a move to stage-2 (orange/very strict) or stage-1 (red/locked down) if and when we reached the specified triggers. In mid-June 2020, Austin’s seven-day rolling average of COVID-19 hospital admissions

surpassed the stage-2 (orange) threshold recommended by our model. In response to this trigger, the mayors of the largest cities in Texas wrote an open letter to the governor, requesting increased local control, which the governor then granted as cases, hospitalizations, and ICU utilization continued to rise. The ongoing tension between state and local leaders motivated the choice of objective functions in this model and has constrained other analyses we have conducted to support the development of *acceptable* policies to save lives while minimizing socioeconomic harms. We have added a short discussion of this to the manuscript.

2. As the referee notes, in our manuscript’s original analysis the penalties started at 1 for the most relaxed stage, and increased by a factor of ten from one stage to the next, but instead used a factor of 100 in going to the strictest stage. Without loss of generality we can select $C_4 = 1$ because only the ratios C_3/C_4 , C_2/C_4 , and C_1/C_4 affect the optimal policy. In principle we understand and agree with the critique, but, as a practical matter, the probabilistic constraint is the primary driver in the model. Given the assumed staging parameters (i.e., estimates for the impact of each stage on transmission), the stage 2 restrictions are not sufficient to reverse rising hospitalizations in late 2020, i.e., the stricter reduction of stage 1 is needed to obey the probabilistic constraint. Our model is effectively “hierarchical.” Roughly speaking with $C_1 \gg C_2$ and $C_2 \gg C_3$, etc. the model first tries to find policies that minimize expected time in stage 1. Among those, it tries to minimize the expected time spend in stage 2, etc. To test the hierarchical hypothesis, we first computed results with $C_1 = 10^3$, $C_2 = 10^2$, $C_3 = 10^1$, and $C_4 = 10^0$. Then we solved the model again with $C_1 = 10^6$, $C_2 = 10^4$, $C_3 = 10^2$, and $C_4 = 10^0$. The obtained optimal thresholds and the detailed results are identical to what was reported in Table 2 in the main text. This shows that the ratio 10 suffices to create such a hierarchical structure for the optimal policy for our problem instances. As a result we now simply use $C_1 = 10^3$, $C_2 = 10^2$, $C_3 = 10^1$, and $C_4 = 10^0$ in the paper. We also now begin Section C of the SI with a discussion of this issue, justifying our choice for the values of the cost parameters, C_1, \dots, C_4 .

Referee #2:

Comments:

The authors present a detailed, data-driven approach to defining triggers for the implementation and relaxation of restriction policies aimed at combating the spread of COVID-19 and maintaining hospital ICU capacity at a level appropriate for treatment of critical cases, during the second wave of COVID-19 in Austin and Houston, TX.

I like this paper, and appreciate the transparent, principled approach the authors took. I am particularly pleased that the authors chose to submit this framework for publication, as it was used to inform policy and therefore will maintain a record of policy decision support methods.

My principal critique is that the method and qualitative results bear strong resemblance to those presented in the precursor paper published in PNAS.

Therefore, this paper would benefit from an explicit discussion of how this work builds upon the previous effort, and specifically how the policy recommendations changed. To be of sufficient novelty for publication in this venue, it is my opinion that some fundamental difference or major scientific development should differentiate the two iterations of the work. Currently it is unclear whether such a difference exists. I say it is unclear because the model, the optimization procedure, and the calibration procedure are all of a nuanced and rigorous nature and for a reader not intimately acquainted with the research, it is difficult to ascertain whether any of the extensions to the proof-of-concept constitute a fundamental step of scientific importance.

The developments in this work are clearly of operational importance to policy design (i.e., replacing ad-hoc parameter estimates with data-driven ones). If the main contribution is the rigorous application of the method described earlier to the second wave of COVID-19 in Austin and Houston, then this relationship between the two pieces of work should be clearly emphasised, and a convincing case made for the importance of that contribution to the scientific literature.

We appreciate this concern and acknowledge that our original submission did not sufficiently distinguish the current work from Duque et al. (2020). We have substantially revised the manuscript and supplement to highlight its novelty and distinct methodological advances, scientific insights and public health applications, including:

1. Actionable policy insights and benchmarks:

The authors of this study have worked together to develop, validate, and consistently follow the staged alert system that has been guiding the COVID-19 response in the City of Austin, Texas since May of 2020; see the Austin Staging Dashboard (Link). Our earlier work in Duque et al. (2020) pre-dated the implementation of this system as well as the summer 2020 COVID-19 surge, both of which substantially changed our understanding of the virus and the challenges faced by city leaderships and healthcare systems. Since Duque et al. (2020), we substantially revised and extended the model to (i) improve fidelity, (ii) derive an actionable policy that achieves the city's health and economic goals and allows consistent public messaging to cultivate community buy-in, and (iii) enable continual validation and rapid analysis of emerging policy challenges (such as triggering the launch of a field hospital). We have revised the manuscript to more fully describe the context and impact of the model on local decision making, public awareness, and epidemiological outcomes. In addition, we explicitly compare Austin's *optimized* triggers to case-count thresholds recommended by Harvard Global Health and ICU-based thresholds used to trigger lock-downs in France, to provide policy insights that are applicable to COVID-19 control in cities worldwide.

2. Methodological advances:

In our current manuscript, the optimization model, i.e., Eqs. [3] in the supporting information (SI) document, provides significant advances relative to Duque et al. to facilitate the analyses and insights referenced above. Rather than toggling between two alert levels as in Duque et al., the current manuscript is based on the five alert stages in use in Austin, and many jurisdictions, and we further extend the model to distinguish ICU capacity, general ward capacity, and surge capacity. (The most relaxed "new normal" stage is not used in our analysis and so we simplify the presentation to four stages.) This required methodological advances. First, we developed a

means to directly estimate parameters governing the impact of control measures from local hospitalization data in Austin and Houston. Second, we developed a new Monte Carlo procedure to infer key model parameters to help ensure our model is consistent with observed hospitalization data. Duque et al. did not analyze hospitalization data to inform parameters for different control measures. These nontrivial methodological advances were essential to deriving, validating, and maintaining a robust multi-staged alert system in Austin since May 2020, and will allow flexible application of the framework for managing future COVID-19 and other pathogen risks in cities worldwide.

3. *Use in practice:*

Since we originally submitted this work for publication in *Nature Communications* in December, the city of Austin has continued to follow the optimized staged alert system described in our manuscript. As prescribed by the policy, Austin declared a transition to the red alert stage when the seven-day average COVID-19 hospitalizations surpassed the specified threshold on December 23, 2020 and relaxed back to the orange state when the admissions declined below the threshold on February 9, 2021. The policy worked as designed. First, the COVID-19 ICU usage peaked just below the local capacity on January 12th. Second, across the 22 Trauma Service Areas of Texas, the Austin region spent the fewest days under state-ordered restrictions on elective surgeries and restaurant/bar/retail occupancy (Executive Order G32). Third, as of March 22, 2021, Texas reported 161 COVID-19 deaths per 100,000 people statewide. In Harris County (Houston), Dallas County (Dallas), Bexar County (San Antonio), the estimated tolls were 120, 143, and 162 COVID-19 deaths per 100,000 people, respectively. Some of our hardest hit regions include the Rio Grande Valley with 281 (Hidalgo County) and 335 (Cameron County) COVID-19 deaths per 100,000 people, and West Texas with 300 (El Paso) and 249 (Lubbock) COVID deaths per 100,000 people. In contrast, Travis County (Austin) reported 73 per 100,000 people during the period, and the five-county Austin MSA reported 77.

We expand on these points in the Appendix below.

In addition to this general critique, I have several more minor questions and comments that I believe could improve the manuscript regardless of where it is published:

1. clarity - currently, both rigorous quantitative, and verbal descriptions of the modelling choices and methods exist in the supplemental material, and I appreciate this greatly. However, clarity would be improved by adjusting the organisation of the supplemental text, tables, and figures, so that the qualitative, verbal descriptions consistently precede the rigorous mathematical descriptions. As a naïve reader, it took quite a while for me to put together all of the pieces, from model, through parameter calibration, optimisation, and sensitivity analysis. I understand that the methods are elaborate, and determining the best presentation is not an easy task, but the current presentation could be improved significantly (though I believe all of the pieces are there).

Thank you for the constructive feedback. We have reorganized the supplement as follows. A new Section A provides an overview of how the subsequent sections, tables, and figures fit together in the context of our overall approach. What is now Section B is the most detailed mathematically, and in that section we now provide verbal descriptions of the transmission dynamics, and of the stochastic optimization model, before the mathematical equations appear.

2. stochastics and scenario rejection - it would be very useful to understand the role of macro- vs micro-stochastic effects in determining the ‘accept’ or ‘reject’ decision described in supplemental section C (selection of scenarios). While I understand that describing this exhaustively may be out of scope, perhaps the overall rejection rate could be provided, along with a rejection rate for fixed macro-stochastics (i.e., those corresponding to the ‘representative trajectory’) where only micro-stochastics produce rejection. I think this particular monte carlo method of forecasting is interesting (I have not encountered it before) and understanding the rejection decision in more detail would be beneficial, as it appears to be useful in providing robust predictions and recommendations.

There are a couple of key factors in the Monte Carlo acceptance-rejection sampling procedure. One is the nature of the stochastics, i.e., macro-stochastic and micro-stochastics. The second is the initial “burn-in” period of February 28–March 23, 2020 (for Austin), which plays an important role, in part because we start with a single infectious individual on February 28th. (Allowing the model to reject scenarios based on the relatively short

initial time period increases the computational efficiency of the Monte Carlo procedure.) In our analysis, we generated two sets of scenarios that are similar to “training” and “testing” data used in statistics and machine learning. We use 300 scenarios for each set, and for Austin this required generating 918,220 scenarios; i.e., the acceptance rate is about 0.07%. Of those scenarios, all but 33,964 were rejected during the initial 25-day period; i.e., conditional on *not* being rejected in the burn-in period, the acceptance rate jumps by a factor of about 25 to 1.8%. In order to assess the effect of micro- and macro-stochastics on our Monte Carlo procedure, we restrict attention to scenarios that have not been rejected during the burn-in period. First, among the scenarios that are ultimately accepted (see Fig. 2 in the supplement), we select the centroid in order to eliminate macro-stochastics. Conditioning on not being rejected during the initial time period, in order to accept 600 scenarios, we must generate 17,030 scenarios meaning the acceptance rate jumps by a factor of about two. Next, we eliminate micro-stochastics by modeling the transitions from one compartment to another based on the expected value, i.e., the deterministic rate associated with the binomial random variables, rather than using Monte Carlo samples from the binomial random variables (which we otherwise use). Again conditioning on not being rejected in the initial 25-day period, the acceptance rate jumps to about 12%. We now provide these details in the supplement.

3. contemporary situation - based on my examination of recent (2nd wave response) events in the Austin area, the thresholds discussed in this piece do not correspond precisely to practice. With the benefit of hindsight, perhaps the authors could add some discussion as to how the situation changed since this work was first produced - which assumptions and modelling decisions have held up, and which have not? This could be added to the discussion ending on line 192 (re: ICU capacity in October).

The main manuscript includes analyses that informed policy in the Austin MSA through September 30, 2020, which includes the summer 2020 surge when the rapid growth in hospitalizations—including critical care patients—caused alarm, and when the city began planning for an alternate care site (ACS) at the convention center to handle lower acuity patients, if needed. (While the beds and equipment were put in place by the end of July, the ACS was in stand-by mode and not staffed—until we hit the specified trigger on January 9, 2021.) Following September, COVID-19 hospitalizations grew relatively slowly in Austin through mid-December and then accelerated to a peak in the middle of January, 2021. Throughout this period and until today (April 2021), the City of Austin has continued to follow the staged alert system that we established in May 2020, enacting and then lifting measures when the seven-day rolling average of COVID-19 hospital admissions crossed the thresholds provided by our analyses. The city transitioned from the yellow to orange alert level on November 19, 2020, and from orange to red on December 23, 2020. The city returned to the orange alert level on February 9, 2021, and then to yellow on March 13, 2021. Throughout this period we have continually refit parameters of our model as new data have come available. We have revised the thresholds as area hospital estimates of ICU capacity have changed. The new SI Section F.4 briefly summarizes analyses we performed to validate and update the staged alert system after September 2020. We also added a discussion in Section 3 of the main text describing our continual *maintenance* of the system through December 2020 and reflecting on lessons learned from the experience.

Overall, this work provides a refreshingly thorough description of what was clearly a rapidly produced system for emergency response. I support publication of the manuscript pending further differentiation from the proof-of-concept piece published by Duque et al. (PNAS, 2020). In addition, any effort to address the minor comments listed above would be appreciated.

Appendix: Contributions of this study relative to our prior work in Duque et al. (2020)

Our earlier work in Duque et al. (2020) pre-dated the May 2020 implementation of the staged alert system in Austin as well as the summer COVID-19 surge, both of which substantially changed our understanding of the virus and the challenges faced by city leaderships and healthcare systems. As a result of our experience in implementing that system and providing decision-support throughout the summer of 2020, the current manuscript contains the following new analyses and policy insights.

- Fundamentally higher-fidelity, more realistic model: The earlier work in Duque et al. was premised on the question policy makers throughout the US were asking in mid-March 2020 of whether to fully lock-down through shelter-in-place orders. The resulting model naively assumed that total hospital space would be the limiting healthcare resource and that there were only two policy options—open or closed. The current work (Yang et al.) differs in three key respects.
 - ★ After the publication of Duque et al., the summer COVID-19 surge clarified that ICU capacity (primarily the ICU workforce) is the limiting resource and that we must account separately for ICU beds, general ward beds, and expanding capacity for low-acuity patients via an alternate care site (ACS). The framework in Yang et al. distinguishes these three forms of capacity, rather than aggregating the former two and ignoring the third;
 - ★ Rather than toggling between *open* and *lock-down*, Austin, like many other jurisdictions, implemented a five-tier policy that allows for more graceful and realistic transition between various stages of social distancing. Yang et al. allow for more than two tiers, both in fitting additional parameters and optimizing a new trigger-based policy.
 - ★ We extended the model to include a ‘fail-safe’ policy, not in Duque et al., that we constructed to provide Austin with triggers for launching an alternate care site (i.e., a field hospital for low acuity COVID-19 patients) that balance the high cost of implementation with the need to ensure sufficient healthcare capacity.
- Key policy insights: By extending the model as just described and then comparing our optimized policies with case count thresholds recommended by Harvard Global Health and ICU-based thresholds used to trigger lock-downs throughout France, we offer the following insights that are broadly applicable to COVID-19 control in cities worldwide.
 - ★ COVID-19 mitigation triggers should explicitly account for ICU capacity, which can be more restrictive than general ward capacity
 - ★ Nonetheless, total COVID-19 hospital admissions (not ICU usage or ICU COVID admissions) is the most reliable key indicator for triggering policy changes among commonly available COVID-19 testing, healthcare, and mortality data.
 - ★ Austin’s stage policy (derived by this framework) is superior to the alternative policies (Harvard and France) in its ability to limit days spent in costly stages of lock-down while reliably staying within ICU capacity.
 - ★ We include an analysis conducted at the request from the Mayor and Public Health Authority of Houston, Texas to demonstrate the flexibility of our approach for other cities.

REVIEWERS' COMMENTS

Reviewer #1 (Remarks to the Author):

I appreciate the authors' earnest efforts in responding to my prior concerns and updating their manuscript accordingly. After reading both files, I feel I now have a better and clearer understanding of the connection between the current manuscript and Duque et al. (2020). It is clear that the paper has advanced Duque et al. (2020) by making the framework more actionable. I also agree with the authors that the paper has made several methodological advances related to how to estimate and infer model parameters. The fact that the proposed model was actually applied in guiding the local decision-making in Austin, Texas was also impressive.

Having said that, I would like to reiterate my prior concern that the paper does not seem to have significantly advanced the frontier of science of infection control. It is true that the framework is more actionable and grounded than that of Duque et al. (2020), but these differences do not necessarily translate into a significant contribution to the academic literature.

The other, perhaps less important, concern is that we are living in May 2021. So the timeliness of the manuscript has significantly diminished. This does not mean the paper does not have value; it is just that the bar should be higher for accepting a paper like this one. Readers living in this age of "warp speed" scientific publications would expect more compelling messages that are potentially applicable to future pandemics.

I am certainly not trying to dismiss the significant practical contribution of this manuscript. It is an insightful and well-written case study and deserves a decent outlet of publication. However, there seems to be a misfit between the journal and the manuscript. I wish the authors best of luck as they continue to look for a home to this interesting work.

Reviewer #2 (Remarks to the Author):

The authors have made substantial revisions to their original draft and have addressed my concerns. In particular, they clarified the significance of this piece with respect to the earlier pilot study already published. I support publication of the current draft.

-Cameron Zachreso

Response to Referees, NCOMMS-20-48848A-Z: Design of COVID-19 Staged Alert Systems to Ensure Healthcare Capacity with Minimal Closures

Referee #1:

Comments:

I appreciate the authors' earnest efforts in responding to my prior concerns and updating their manuscript accordingly. After reading both files, I feel I now have a better and clearer understanding of the connection between the current manuscript and Duque et al. (2020). It is clear that the paper has advanced Duque et al. (2020) by making the framework more actionable. I also agree with the authors that the paper has made several methodological advances related to how to estimate and infer model parameters. The fact that the proposed model was actually applied in guiding the local decision-making in Austin, Texas was also impressive.

Having said that, I would like to reiterate my prior concern that the paper does not seem to have significantly advanced the frontier of science of infection control. It is true that the framework is more actionable and grounded than that of Duque et al. (2020), but these differences do not necessarily translate into a significant contribution to the academic literature.

The other, perhaps less important, concern is that we are living in May 2021. So the timeliness of the manuscript has significantly diminished. This does not mean the paper does not have value; it is just that the bar should be higher for accepting a paper like this one. Readers living in this age of "warp speed" scientific publications would expect more compelling messages that are potentially applicable to future pandemics.

I am certainly not trying to dismiss the significant practical contribution of this manuscript. It is an insightful and well-written case study and deserves a decent outlet of publication. However, there seems to be a misfit between the journal and the manuscript. I wish the authors best of luck as they continue to look for a home to this interesting work.

We thank the referee for the constructive comments.

Staged alert systems have been a key component of the global COVID response. However, we utterly lack an evidence base for the rapid design and deployment of such policies in the face of novel threats. Our study is one of the first to tackle this challenge and to systematically evaluate several of the staged systems that have governed COVID actions worldwide. In our opinion, our article not only provides fundamental insights into the design of adaptive (data-triggered) policies to manage a non-linear and highly uncertain pathogen threat, but also highlights a critical new direction for interdisciplinary scientific inquiry at the interface of epidemiology, behavior, and policy.

We are still a long way from eradicating COVID and will likely face novel pandemic threats in the decades ahead. We agree that, as vaccinations continue to roll out, the need for non-pharmacological interventions and lock-downs will diminish. However, the pace of vaccination across much of the world lags far behind the US and UK and the continual emergence of new variants may lead to future resurgences even in highly vaccinated communities. Thus, staged alert systems will continue to be an important option for mitigating risks, particularly if they are carefully tailored to the changing state of the pandemic and effectively communicated to encourage public compliance. In Austin, for example, more recent analyses suggest that we cannot yet abandon the staged alert system without running the risk of unmanageable hospital surges, even though 40% of the population has been fully vaccinated. We are applying these methods to design updated policies in an era of increasing vaccine coverage and possible seasonal emergence of variants.

Thus, we believe that our article offers more than just a case study. We provide a rigorous approach for retrospectively assessing COVID policies, as a key component of intra-action and after-action reviews, like the recent report by the

WHO's Independent Panel for Pandemic Preparedness and Response. Equally important, we provide a flexible framework for designing policies to navigate the coming epidemic and potentially endemic phases of COVID and future pandemic threats.

We have added a brief comment highlighting these contributions in the final paragraph of the paper's Discussion section.

Referee #2:

Comments:

The authors have made substantial revisions to their original draft and have addressed my concerns. In particular, they clarified the significance of this piece with respect to the earlier pilot study already published. I support publication of the current draft.

We thank the referee for reviewing the manuscript.